# Transcriptome-Based Analysis Reveals a Crucial Role of *BxGPCR17454* in Low Temperature Response of Pine Wood Nematode (*Bursaphelenchus xylophilus*)

**DOI:** 10.3390/ijms20122898

**Published:** 2019-06-14

**Authors:** Bowen Wang, Xin Hao, Jiayao Xu, Yan Ma, Ling Ma

**Affiliations:** 1College of Forestry, Northeast Forestry University, Harbin 150040, China; wangbowen6@126.com (B.W.); xinhao@nefu.edu.cn (X.H.); guooguoo@126.com (J.X.); 2College of Management, Harbin University of Commerce, Harbin 150028, China

**Keywords:** *Bursaphelenchus xylophilus*, pine wilt disease, low temperature, transcriptome, GPCR, RNAi

## Abstract

Background: The causal agent of pine wilt disease is the pine wood nematode (PWN) (*Bursaphelenchus xylophilus*), whose ability to adapt different ecological niches is a crucial determinant of their invasion to colder regions. To discover the molecular mechanism of low temperature response mechanism, we attempted to study the molecular response patterns under low temperature from *B. xylophilus* with a comprehensive RNA sequencing analysis and validated the differentially expressed genes (DEGs) with quantitative real-time polymerase chain reaction (qRT-PCR). Bioinformatic software was utilized to isolate and identify the low-temperature-related BxGPCR genes. Transcript abundance of six low-temperature-related BxGPCR genes and function of one of the BxGPCR genes are studied by qRT-PCR and RNA interference. Results: The results showed that we detected 432 DEGs through RNA sequencing between low-temperature-treated and ambient-temperature-treated groups nematodes. The transcript level of 6 low-temperature-related BxGPCR genes increased at low temperature. And, the survival rates of *BxGPCR17454* silenced *B. xylophilus* revealed a significant decrease at low temperature. Conclusion: in conclusion, this transcriptome-based study revealed a crucial role of *BxGPCR17454* in low temperature response process of pine wood nematode. These discoveries would assist the development of management and methods for efficient control of this devastating pine tree pest.

## 1. Introduction

As one of the most dangerous plant pests in the world, pine wood nematode (*Bursaphelenchus xylophilus*, PWN) causes devastating pine wilt diseases to the pine trees in Asia, Europe, and North America [1]. Unfortunately, the infestation area of PWN, in all probability, will continue expanding to colder regions of Asia and Europe [2,3,4,5]. For many parasites, PWN included, their ability to adapt different ecological niches is a crucial determinant of their invasion to new regions [6]. The formation of low-temperature-induced diapause stage of PWN is one of the most pivotal factors for PWN’s survival and expansion. In the autumn, PWN will gradually stop their growth and gradually enter winter diapause in response to the decreased environmental temperature. Additionally, the second-stage propagative juveniles (J_2_) gradually turn into specialized third-stage dauer larva (DL_3_), so that they can enter winter diapause together with the adults [7,8,9]. All the low-temperature-induced nematodes revealed similar characteristics: extended lifespan and accumulated fat content [8]. 

Although very little is known about the low temperature-sensing mechanism in PWN, much effort has been invested in probing the functions of some genes that respond to the low temperature in the model nematode *Caenorhabditis elegans*. The low temperature response process of *C. elegans* is regulated by many genes and groups including Cyclic guanosine monophosphate (cGMP) pathway, insulin-signaling pathway [10], G protein signaling [11], and phospholipid saturation [12], among others. The analysis of this widely studied model nematode inspired our analysis for the mechanism study of low temperature sensation in PWN. Some efforts have been made in the study of molecular mechanism of PWN’s response to low temperature in our previous research: we proved the key role of patched-related protein gene *Bx-DAF6* [13], cGMP pathway [14], and stearoyl-coA desaturase gene *Bx-SCD* [15]. However, the mechanism of low-temperature-induced lifespan extension of PWN remains elusive. 

In this study, we performed RNA sequencing on mixed-stage PWN between ambient temperature treated and low temperature treated groups. The results of RNA sequencing illustrated an essential role of G protein signaling during PWN’s early response to the low temperature. G protein coupled receptors (GPCRs) are a functionally diverse superfamily which are known as widely related to the environmental signal detection process [16,17,18]. Thus, we identified six GPCRs genes in G protein signaling from PWN and studied their response pattern to low temperature. We also validated the function of one GPCRs gene in low temperature-sensing with RNAi method. The results reveal that *BxGPCR17454* plays a key role in PWN’s response to low temperature. This means the *BxGPCR17454* in PWN may have the potential to be treated as a target for the management of this dangerous pest.

## 2. Results 

### 2.1. RNA Sequencing and General Transcription Patterns 

We performed RNA sequencing and obtained 312,138,316 high quality clean reads in all six samples after removal of rRNA and low-quality reads (Table 1). The read counts were converted to FPKM. Dendrogram clustering of the biological duplicates for each RNA sequencing condition reveals conserved alignment between replicates. (Figure 1A). The proportion of total reads that mapped to the reference genome ranged from 70.47% to 75.61%, and correlation values were significantly higher between the duplicated samples than among the treatments (Figure 1B).

### 2.2. B. xylophilus Genes Differentially Expressed in Response to Low Temperature

Between low-temperature-treated and CK nematodes, we identified 432 DEGs of which 314 were up-regulated, while 118 DEGs were down-regulated (Figure 2A, Appendix A). Among all the DEGs, the gene expression patterns were similar between the biological duplicates but significantly different between the low-temperature-treated and CK group. (Figure 2B). GO enrichment showed that the DEGs mainly distributed in the single-organism process, cellular process, metabolic process, developmental process, and response to stimulus in biological process terms and binding and catalytic activity in molecular function terms (Figure 2C). Our results also indicated that low temperature influenced the expression of several gene groups with the putative function of environmental signal detection and stress adaptation (Appendix A). These genes groups include four cytochrome P450s (BXY_0803400, BXY_0817800, BXY_1185500 and BXY_1697600), six heat shock proteins (BXY_0165400, BXY_0586000, BXY_0640100, BXY_0768000, BXY_1274600 and BXY_1563600) and 16 GPCRs (Appendix A), among others. 

### 2.3. Validation of DEGs by Quantitative Real-Time Polymerase Chain Reaction (qRT-PCR)

12 DEGs were selected randomly for qRT-PCR validation. In general, the patterns of up-regulation and down-regulation were consistent with those obtained from the RNA sequencing analysis. Thus, qRT-PCR analysis confirmed that the changes detected in the RNA sequencing analysis were reliable (Figure 3, Appendix A). 

### 2.4. Identification and Transcript Abundance Analysis of 6 Low-temperature-related BxGPCRs

GPCRs are a group of genes which has been reported as closely related to the environmental signal detection process [16,17,18]. For this reason, we studied the relationship of low temperature stress and BxGPCRs expression. Structurally, GPCRs are also known as 7TM receptors because they have seven transmembrane (TM) domains [18,19]. Thus, we then used conserved domain analysis and transmembrane region prediction to filter the 16 differently expressed genes with GPCRs putative function description. BxGPCRs with incomplete conserved domain (Appendix A) or with numbers of TM not equal to seven (Appendix A) are not further considered. There are six low-temperature-related BxGPCRs left after this filtering. According to these six genes’ ID BXY_1375800, BXY_1672700, BXY_0593200, BXY_0521000, BXY_1391500, and BXY_1745400, we named them BxGPCR13758, BxGPCR16727, BxGPCR05932, BxGPCR05210, BxGPCR13915, and BxGPCR17454, respectively. Blastp results showed that the deduced amino acid sequence of these six low-temperature-related BxGPCRs have a relatively high level of identity with other GPCR proteins of several nematodes. On this basis, we downloaded selected homologous amino acid sequences from NCBI (Appendix A), performed alignment of the deduced amino acid sequences (Appendix A) and constructed phylogenetic tree (Figure 4A). 

To validate the transcript pattern of BxGPCRs under low temperature, we measured transcript levels of six BxGPCRs at 5 °C for 1, 3, 5, and 7 days, and 25 °C for 1, 3, 5, and 7 days, respectively, with qRT-PCR (Figure 4B). The result showed that all of 6 BxGPCRs genes revealed higher transcript levels at low temperature than regular temperature over 7 days. The gene expression culminates at 3 days. 

### 2.5. RNAi Validation of Low-temperature-related BxGPCR

The FITC treated nematodes reflected green fluorescence after exposed under ultraviolet light (Figure 5A), which indicated that the dsRNA can be fully absorbed by nematodes with soaking method. According to the result of expression changes between BxGPCR17454 dsRNA-treated and dsRNA-free nematodes, BxGPCR17454 gene was significantly silenced after dsRNA treatment. The mean expression level of BxGPCR17454 was decreased to 11.85% compared to the dsRNA-free nematodes. However, the dsRNA had no obvious effect on the transcript level of another internal control gene β-actin (Figure 5B). The alignment of six BxGPCRs nucleic sequences also illustrated that RNAi primer sequences is unique for BxGPCR17454 (Appendix A). These results indicated that the BxGPCR17454 RNAi was potent and specific in this study. 

The results of the survival rates calculation showed that *B. xylophilus* revealed a significantly decreased survival rate after RNAi of BxGPCR17454 under low temperature compared to dsRNA-free group over 30 days (Figure 5C). This indicated that gene silencing of BxGPCR17454 can significantly decrease the survival rate of *B. xylophilus* at low temperature. BxGPCRs plays a key role in the low temperature response process of *B. xylophilus*. Although the transcript level of BxGPCR17454 was evaluated in the first 7 days according to the qRT-PCR results, there are no differences between the survival rate of dsRNA-free group and dsRNA-treated groups in the first 15 days. This may be because there are other downstream genes are influenced by the RNAi of BxGPCR17454. All these genes together with BxGPCR17454 influenced the survival rate of *B. xylophilus* under low temperature. Further studies need to be carried out to detect the downstream genes of BxGPCR17454 in the future. 

## 3. Discussion

It is commonly known that low temperature response processes of both homeotherms and poikilotherms are not passive thermodynamic process but active genetic-promoted processes [20]. In this study, our results coincided with this theory. Transcriptome analysis revealed that low temperature response process of this plant parasite nematode is also a genetic process regulated by many genes. The past 15 years has witnessed a rapid progress in the understanding of how low temperature influenced the genetic response patterns of many plant pests with RNA sequencing method [21,22,23,24,25,26]. However, little is known about how *B. xylophilus*, which is also one of the most destructive pine tree pests, genetically responded to low temperature. 

The identification and characterization of genes involved in low temperature responses is essential to elucidate low temperature defense mechanisms and develop effective control strategies. Changes in the transcriptome of *B. xylophilus* during low temperature directly reflect the impact of low temperature on the genetic activities. In this article, we conducted a comprehensive transcriptome analysis and characterized the gene expression profiles of *B. xylophilus* under low temperatures. Through the analysis of DEGs sets, we identified transcriptome changes in *B. xylophilus* in response to low temperature and validated the transcriptome result with qRT-PCR. The DEG analysis revealed that many gene groups including heat shock proteins, cytochrome P450s and GPCRs, among others, may potentially have played a crucial role in low temperature response process of *B. xylophilus*. On this basis, six low-temperature-related BxGPCRs genes were identified by phylogenetic analysis. In addition, we found these six low-temperature-related BxGPCRs genes revealed higher transcript abundance under low temperature than ambient temperature. Furthermore, the RNAi method was utilized to study the functions of one of the low-temperature-related BxGPCRs which we named *BxGPCR17454.* The results indicated that interference of *BxGPCR17454* can significantly reduce the survival rate of *B. xylophilus* under low temperature. Taking all these factors into consideration, we hypothesized that G protein signaling may play a crucial role for the low temperature response process in *B. xylophilus.* This is widely consistent with previous research results in other organisms such as nematodes [10,11,27], bacterium [28], plants [29,30], fish [31], and mammals [32,33,34]. Interference of *BxGPCR17451* may represent a novel approach for the prevention of this dangerous plant pest. Apart from G protein signaling, others DEGs such as heat shock proteins, are widely investigated as a big gene group regarding their role in thermal stress response, development and lifespan regulation [35,36,37,38]. Cytochrome P450s are not only reported as an import detoxification gene group [39], but also thermal-stress-related genes [40,41]. Further validations are expected to be made in these gene groups in the future. Other gene groups and pathways that have been reported to have a low temperature response function in model organism *C. elegans* include TRP channels [20,42,43], the cAMP pathway [44], and fatty acid metabolism pathway [45,46]; their orthologous genes in *B. xylophilus* may also share similar low temperature response functions. Further efforts are also expected to be made in the relevant fields in the future. 

This study focused on the molecular response pattern of *B. xylophilus* under low temperature, identification and response pattern validation of six low-temperature-related BxGPCRs genes as well as functional validation of *BxGPCR17454*. The results shaped an important role of a G protein signaling gene *BxGPCR17454* and potential value of other low-temperature-related genes in the low temperature response process of *B. xylophilus.* These discoveries would assist the development of management and methods for efficient control of this pine tree pest.

## 4. Material and Methods

### 4.1. Sample Preparation

*B. xylophilus*, maintained in the Forestry Protection Laboratory of Northeast Forestry University, Harbin, China, were kindly provided by the Chinese Academy of Forestry, Beijing, China. The nematodes were cultured on *Botrytis cinerea* for 5–7 days at 25 °C. The Baermann funnel technique was used to isolate *B.xylophilus* from potato dextrose agar medium plates. About 60,000 nematodes were mixed evenly in 6 mL distilled water. Then, suspensions were separated into six 1.5 mL centrifuge tubes equally. Three tubes were cultured at 5 °C for 24 hours as treated group. The other three tubes were cultured at 25 °C for 24 hours as CK group. After the incubation was finished, all samples were collected in the bottom of the tubes with centrifuge. Then, the tubes were all transferred in liquid nitrogen immediately for the next analysis.

### 4.2. RNA Sequencing

Total RNA of above six samples was extracted with Trizol reagent (Thermo Fisher Scientific, Shanghai, China). RNA-seq library was then constructed with NEB #7530 Kit (New England Biolabs, Ipswich, MA, USA) as following method: mRNA was enriched by Oligo(dT) beads. First, the enriched mRNA was fragmented into short fragments using NEBNext First Strand Synthesis Reaction Buffer. Random primers together with ProtoScript Ⅱ Reverse Transcriptase and Marine RNase Inhibitor were then used to reverse transcript mRNA into cDNA with following procedure: 25 °C 10 min, 42 °C 15 min, 70 °C 15 min. The products were then added with Second Strand Synthesis Reaction Buffer and Second Strand Synthesis Enzyme Mix to synthesis second-strand cDNA under 16 °C 60 min. Next, the cDNA fragments were purified with 1.8X Agencourt AMPure XP Beads. Sequencing adapters were then ligated with purified second-strand cDNA with follow steps: first, purified double stranded cDNA was pre-treated with NEBNext End Repair Reaction Buffer and NEBNext End Prep Enzyme Mix under 20 °C 30 min, 65 °C30 min. Then, adapters were added in the pre-treated second-strand cDNA with Blunt/TA Ligase Master Mix and Diluted NEBNext Adaptor under 20 °C 15 min. The ligation products were size selected by agarose gel electrophoresis, PCR amplified with 12-15 cycles, and sequenced using Illumina HiSeqTM 2500. RNA extraction, RNA-seq library construction and sequencing are performed by Gene Denovo Biotechnology Co. (Guangzhou, China). 

### 4.3. Sequencing Data Analysis

Raw reads containing adapters, containing more than 10% of unknown nucleotides or containing more than 50% of low quality (Q-value ≤ 20) bases were removed by fastp [47] (Plant Pathology, SCRI, Invergowrie, Dundee DD2 5DA, UK. version 0.18.0) in order to get high quality clean reads. Short reads alignment tool Bowtie2 [48] was used for mapping reads to ribosome RNA (rRNA) database with default parameters. The rRNA mapped reads were removed. The remaining rRNA removed reads of each sample were then mapped to reference genome by TopHat2 [49] (Center for Bioinformatics and Computational Biology, University of Maryland, College Park, MD, USA. version 2.0.3.12). The mapping options used with TopHat are as follows: maximum read mismatch is 2, the distance between mate-pair reads is 50bp and the error of distance between mate-pair reads is ±80bp. The reference genome of *B. xylophilus* which was submitted directly to WormBase database by Taisei Kikuchi in 2011 [7] was downloaded from WormBase Parasite (BioProject PRJEA64437). The reference genome has 17,704 transcripts in total. We used RSEM software [50] (Department of Computer Sciences, University of Wisconsin-Madison, Madison, WI, USA) to calculate the gene expression level which was normalized by using FPKM (Fragments Per Kilobase of transcript per Million mapped reads) method. The edgeR package (Bioconductor, Roswell Park Cancer Institute, Buffalo NY, USA. http://www.r-project.org/) was used to identify differentially expressed genes with raw counts from RSEM across two groups. We identified genes with a fold change ≥2 and a false discovery rate (FDR) <0.05 in a comparison as significant differentially expressed genes (DEGs). BLASTX against the NCBInr database and Gene Ontology database was used to analysis gene description and annotations. Clustering, Pearson correlation analysis, volcano plot, heatmap, and GO categories are performed by R (Bioconductor, Roswell Park Cancer Institute, Buffalo NY, USA. version 3.2.1). 

### 4.4. Validation of DEGs by Quantitative Real-Time Polymerase Chain Reaction (qRT-PCR)

Quantitative real-time Polymerase Chain Reaction (qRT-PCR) was performed to validate the expression of DEGs. Gene-specific primers (Appendix A) for 12 randomly selected DEGs were designed by Sangon Online Primer Designer (available online: https://www.sangon.com/newPrimerDesign) (Sangon Biotech Co. Shanghai, China). The *Bx28S* gene was used as the internal control gene (Appendix A). Double-stranded cDNA was obtained using GoTaq 2-Step RT-qPCR System (Promega, Madison, WI, USA) according to instructions of the manufacturer. We used the cycle threshold data to calculate the fold of relative transcript abundance (treated group/CK group). The PCR program used in this analysis was first step: 95 for 2 min, second step: 95 for 15 s and 60 for 1 min, in 40 cycles. 

### 4.5. Identification of Low-temperature-related BxGPCRs

According to the analyzed sequencing data, DEGs with a NR description of GPCRs were selected as candidate low-temperature-related *BxGPCRs*. Candidate genes with incomplete conserve domain were removed. The candidate genes whose predicted transmembrane domains unequal to seven were also removed, considering seven transmembrane domains is a typical feature of GPCRs [51,52]. Conserved domains were analyzed by NCBI Batch CD (available online: https://www.ncbi.nlm.nih.gov/Structure/bwrpsb/bwrpsb.cgi) and transmembrane domains were predicted by the online tool TMHMM from the Center for Biological Sequence website (available online: http://www.cbs.dtu.dk/services/TMHMM/). *BxGPCRs* homologous sequences of the other organisms were obtained from BLASTP (available online: https://blast.ncbi.nlm.nih.gov/Blast.cgi). Alignments of deduced amino acid sequences were performed by Muscle (EBI, UK) [53] and visualized by Jalview (version 2. 10. 4b 1). Phylogenetic analyses were performed with Mega 7.0 software using a maximum likelihood tree. 

### 4.6. Transcript Abundance Analysis of 6 BxGPCRs under Low Temperature 

The transcript abundance was analyzed as the method described before [14]. About 80,000 nematodes were mixed evenly in 8 mL distilled water. Then we separated the suspensions into eight 1.5 mL centrifuge tubes equally. Four tubes of nematodes were cultured at 5 °C for 1, 3, 5, and 7 days. The other four tubes of nematodes were cultured at 25 °C for 1, 3, 5, and 7 days. Total RNA of the nematodes cultured in these eight tubes were extracted and synthesized into cDNA with the method as described above after the incubation was finished. The transcript abundance and qRT-PCR Primers (Appendix A) were also calculated and designed as described above. The whole transcript abundance analysis was analyzed with three replicates as three independent trials. 

### 4.7. RNAi Validation of Low-Temperature-Related BxGPCR

We utilized RNAi method to study the function of one of the low-temperature-related BxGPCRs *BxGPCR17454.* The nematode RNAi was performed with soaking method as outlined in previous papers [14,39,54,55,56]. Mixed-stage *B. xylophilus* (male: female: juvenile ratio is approximately 1:1:2) was used in this research. MAXIscript T7/T3 RNA Synthesis Kit (Ambion, Tokyo, Japan) was used to obtain dsRNA with the following primers: i-BxGPCR17454-F (GCT AAT ACG ACT CAC TAT AGG GGG AAT CAT TGT ATA CCA GCC AAT TGC) and i-BxGPCR17454-R (AGT AAT ACG ACT CAC TAT AGG GTG GCC CGA TTG AAA TCC TGC). The size of *BxGPCR17454* dsRNA for RNAi is 350 bp. The dsRNA-treated nematodes were soaked in ddH_2_O with final concentration of 2 mg/mL dsRNA corresponding to *BxGPCR17454* sequence for 24 hours at 25 °C to uptake the dsRNA. The uptake of the dsRNA was monitored by a final nematode treatment of ddH_2_O containing 1 mg/mL FITC (fluorescein isothiocyanate) for 24 hours. The CK nematodes were soaked in ddH_2_O only. After intermittent stirring for 24 hours at 25 °C, the nematodes were washed with ddH_2_O to remove the external FITC and dsRNA. A fluorescence microscope was then used to take pictures of the nematode (ZEISS, Germany) to detect the uptake of FITC. Nematodes were divided into two groups after dsRNA uptake: the first group was used to examine the efficiency of RNAi with qPCR, and the second one was used to calculate survival rate under 5 °C environments. Total RNA was extracted from the CK nematodes and the dsRNA-treated nematodes after dsRNA. qRT-PCR was then performed as mentioned above with qRT-PCR primers of *BxGPCR17454* and internal control primers *Bx28S* and *β-actin* [57] (Appendix A). Survival rates of nematodes were calculated with 100 nematodes in 1 mL of ddH_2_O within each 1.5 mL centrifuge tube performed in triplicate. Then the survival rate of nematodes in each centrifuge tube was monitored every 3 days over 30 days. The survival rates of each tube were calculated as follows: number of living nematodes in each tube / number of total nematodes in each tube * 100%.

## Figures and Tables

**Figure 1 ijms-20-02898-f001:**
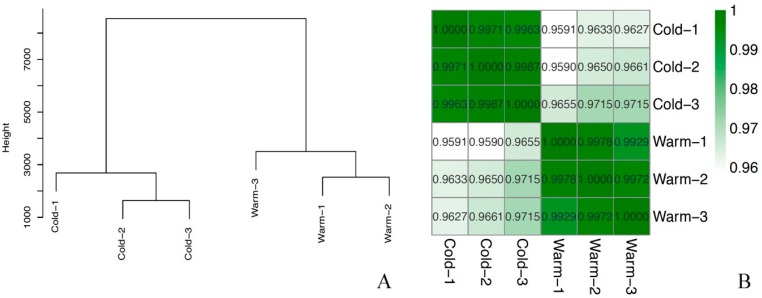
General transcription patterns of all samples (**A**). Clustering of the biological duplicates for each RNA sequencing condition reveals conserved alignment between replicates. (**B**). Pearson correlation between low-temperature-treated and CK nematodes.

**Figure 2 ijms-20-02898-f002:**
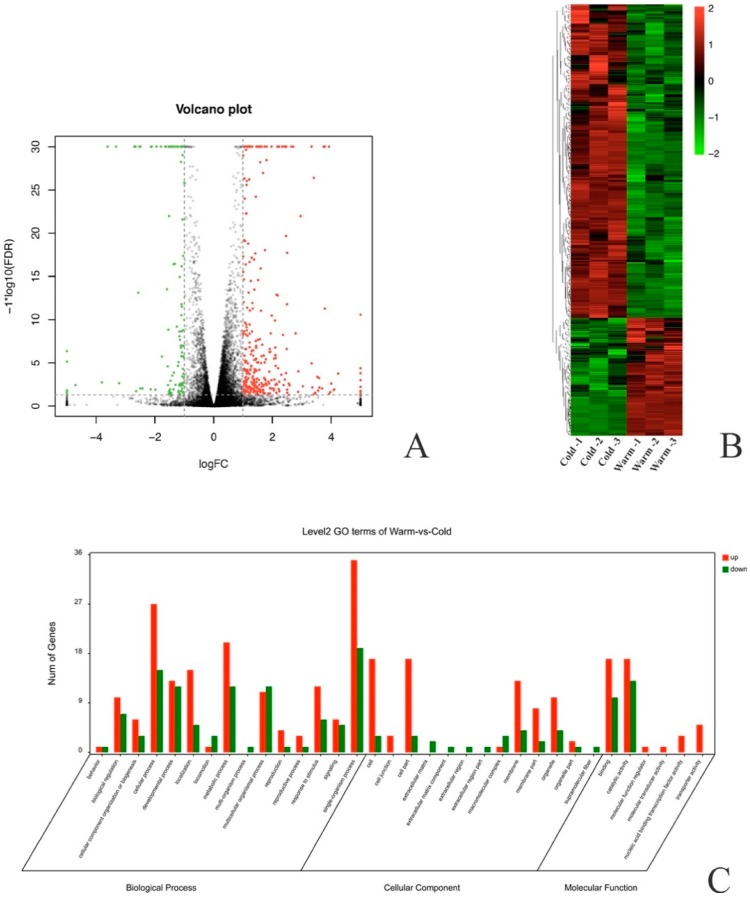
Differentially expressed genes (DEGs) analysis. (**A**). DEGs between low-temperature-treated and CK group nematodes. Red dots represent up-regulated genes and green dots represent down-regulated genes. (**B**). Heat map of all the DEGs between low-temperature-treated and CK group nematodes. (**C**). Gene Ontology (GO) categories of DEGs between low-temperature-treated and CK group nematodes.

**Figure 3 ijms-20-02898-f003:**
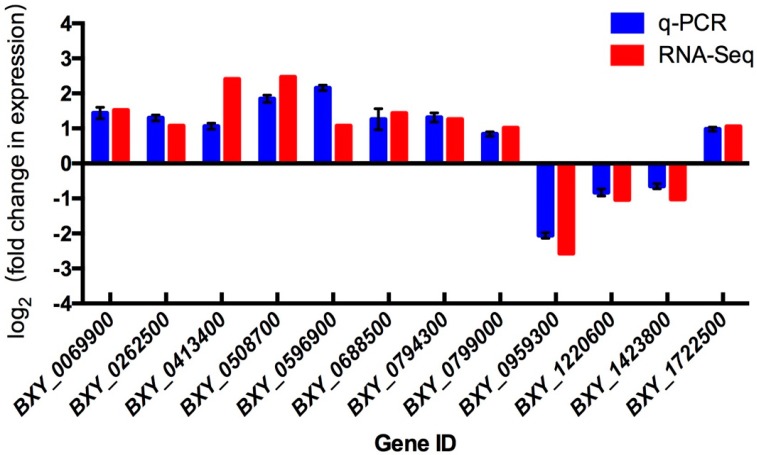
Quantitative real-time polymerase chain reaction (qRT-PCR) identification of differentially DEGs between low-temperature-treated and CK group nematodes. The relative gene expressions were normalized with the internal control gene Bx28S. The putative function of each gene is as follows: BXY_0069900, NH127; BXY_0262500, SERA; BXY_0413400, ST2B1; BXY_0508700, LIP1; BXY_0596900, ASK12; BXY_0688500, NHR54; BXY_0794300, NPY2R; BXY_0799000, DUS9; BXY_0959300, ADH3; BXY_1220600, SOCS4; BXY_1423800, AATC; BXY_1722500, CYSK2.

**Figure 4 ijms-20-02898-f004:**
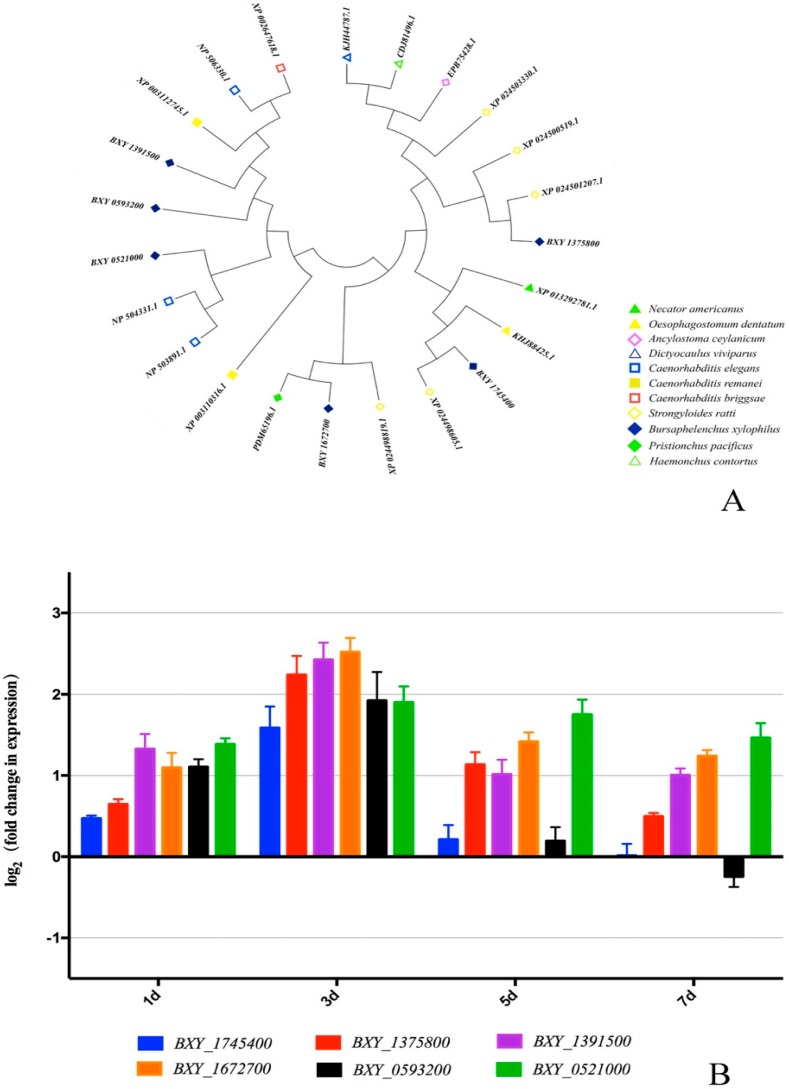
Identification and transcript pattern analysis of six low-temperature-related BxGPCRs. (**A**). Analysis on phylogenetic trees of six deduced protein sequences with other organisms. Labels at the end of each branch are gene id/NCBI accessions of each gene. Different shapes stand for the Latin name of each organism. (**B**). Transcript abundance analysis of six BxGPCRs under low temperature.

**Figure 5 ijms-20-02898-f005:**
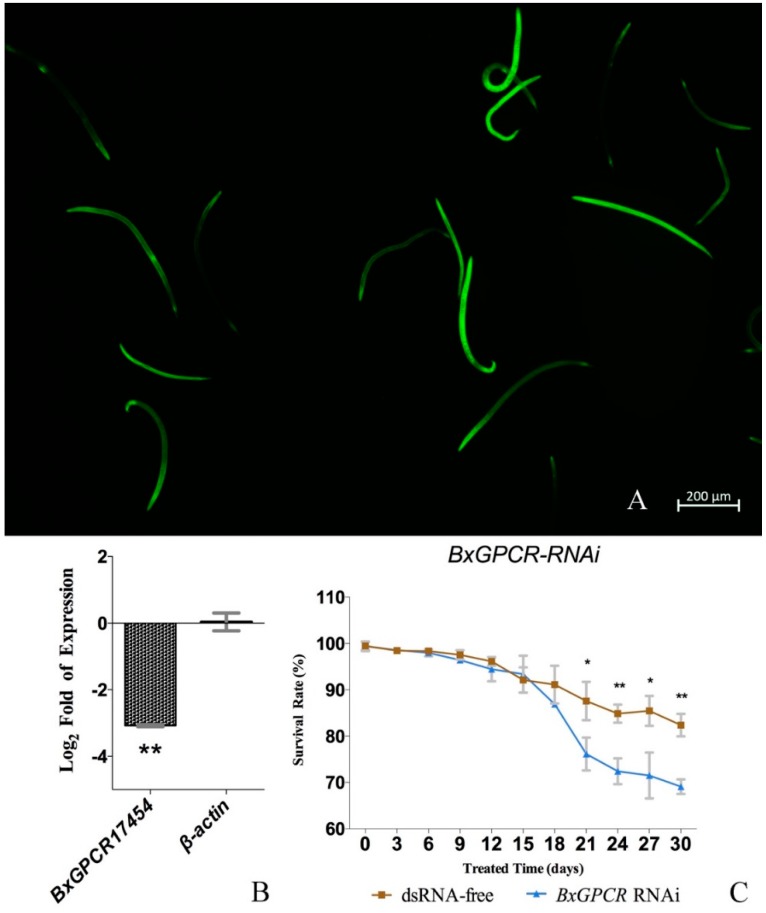
RNAi validaton of BxGPCR17454. (**A**). *B. xylophilus* soaked in fluorescein isothiocyanate (FITC) revealed green fluorescent signal under ultraviolet light. (**B**). BxGPCR17454 RNAi efficiency was validated by expression changes between BxGPCR17454 dsRNA-treated and dsRNA-free nematodes. BxGPCR17454 gene was significant silenced after dsRNA treatment, while dsRNA had no obvious effect on the transcript level of internal control gene β-actin. (**C**). BxGPCR17454 dsRNA-treated nematodes showed significate decreased survival rate under low temperature. Data represent mean values ± SD from different repetitions. Asterisks indicates statistically significant differences (* *p* < 0.05, ** *p* < 0.01, Student’s t-test) was found between the dsRNA-treated and dsRNA-free groups.

**Table 1 ijms-20-02898-t001:** Statistics analysis of RNA sequencing data from low-temperature-treated and CK nematodes.

Samples	Clean Reads	High Quality Clean Reads (%)	High Quality Clean Bases	Q20 (%)	Q30 (%)	GC (%)	Mapping Rate
Cold-1	51,107,560	47,519,530 (92.98%)	6,850,534,848	95.08%	86.34%	48.70%	75.24%
Cold-2	47,251,708	43,777,894 (92.65%)	6,230,763,134	94.72%	85.36%	48.35%	73.29%
Cold-3	43,275,876	40,225,612 (92.95%)	5,760,600,645	94.92%	85.93%	48.62%	75.36%
Warm-1	46,243,122	42,555,170 (92.02%)	6,101,120,213	94.74%	85.43%	48.87%	75.61%
Warm-2	45,717,826	42,541,694 (93.05%)	6,065,096,300	94.87%	85.83%	48.74%	75.56%
Warm-3	98,615,884	88,413,574 (89.65%)	12,105,837,900	93.42%	82.30%	48.27%	70.47%
**Total**	332,211,976	305,033,474	43,113,953,040				

Notes: Cold-1, Cold-2 and Cold-3 stands for low-temperature-treated nematodes sample 1, 2, and 3. Warm-1, Warm-2, and Warm-3 stands for CK group nematodes sample1, 2, and 3. Q20 stands for the percentage of bases with a Phred value >20. Q30 stands for the percentage of bases with a Phred value >30.

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
