# Peer review of "Transcriptome-Based Analysis Reveals a Crucial Role of *BxGPCR17454* in Low Temperature Response of Pine Wood Nematode (*Bursaphelenchus xylophilus*)"

_ijms, 2019, doi:10.3390/ijms20122898_

Round 1

Reviewer 1 Report

The manuscript entitled "Transcriptome-based Analysis Reveals a Crucial Role of BxGPCRs in Low Temperature Response of Pine Wood Nematode (Bursaphelenchus xylophilus)" has been submitted by Bowen Wang et al. On the whole, the study appears to have been carefully performed. However, I’m not convinced by the term “crucial role of BxGPCRs”: just one gene was tested for its putative role on the nematode survival rate (70% survival rate when the gene is silenced instead of 85% for control nematodes): authors should change the title and modify the discussion section in order to be less affirmative.

Overall, the manuscript must be rewritten: there are grammatical errors, incomplete sentences…, the results section is too succinct and need improvement and supplemental data

M&M:

2.2 Indicate the company for each enzyme or kit

2.3 Indicate the mapping options used with TopHAt

2.6 “The other four tubes of nematodes were cultured at 5 for 1, 3, 5 and 7 days.” Same temperature?

“One tube of nematodes was used to extract the RNA and synthesize that RNA into cDNA directly without treatment (no treated group).” Day 0?

2.7 Mixed-stages Bx: which ones?

 How do you calculate survival rate?

Results:

3.1 Too succinct: “we performed RNA sequencing” of what? Please, describe the samples, the mapping step, include the percentage of mapping in Table 1 and the number of transcripts detected for each sample (indicate somewhere the total number of transcript for the Bx reference)

3.2 GO enrichment is not very informative (too general), could you describe some DEG (just to highlight 2 or 3 categories)

3.3 Figure 3: please, indicate in the legend the (putative) function of each gene

3.4 Add a figure showing alignment of the deduced amino acid sequences from the 6 predicted BxGPCR, and a known GPCR (from C. elegans), indicate the 7 transmembrane domains.

What are the percentage of identity between the 6 BxGPCR and with other GPCRs?

For QPCR analyses, indicate that the gene expression culminates at 3 days.

3.5 What are the homologies between the 6 BxGPCR nucleic sequences? I wonder if a dsRNA of 350 b is specific or can target another GPCR? Did you test the expression level of other GPCRs?

“The BxGPCR17454 RNAi was potent and specific for B. xylophilus.” What does it mean?

Discussion

-“We proved this theory in B. xylophilus through transcriptome analysis” too affirmative!

-Please include a brief description of GPCR: structure, how many genes have been described in C. elegans, …

-discussion on other DEG? HSP or other genes known to be related to thermal stress?

Author Response

Dear Reviewer,

Thank you so much for the reviewers’ comments concerning our manuscript. Those comments are all valuable and very helpful for revising and improving our paper, as well as the important guiding significance to our research. We have studied comments carefully and have made correction. We tried our best to improve the manuscript and made some changes in the manuscript. Revised portion are marked with "Track Changes" function in Microsoft Word in the paper.

With the help of your kind comments, we now understand that only one gene is not enough to prove “the crucial role of BxGPCRs”. So, we revised the title and paper topic have been revised into “Transcriptome-based Analysis Reveals a Crucial Role of BxGPCR17454in Low Temperature Response of Pine Wood Nematode (Bursaphelenchus xylophilus)”. 

We appreciate for reviewers’ warm work earnestly, and hope that the correction will meet with approval. Once again, thank you very much for your comments and suggestions.

Best regards,

Bowen Wang

Author's Reply to the Review Report (Reviewer 1)

M&M:

2.2 Indicate the company for each enzyme or kit

Reply: Thanks. The kit we used here is a commercial Kit by New England Biolabs. We have added the details in the revised manuscript. 

2.3 Indicate the mapping options used with TopHat

Reply: Thanks. The mapping options used with TopHat are as follows: maximum read mismatch is 2, the distance between mate-pair reads is 50bp and the error of distance between mate-pair reads is ±80bp. We have added these details in the revised manuscript.

2.6 “The other four tubes of nematodes were cultured at 5 for 1, 3, 5 and 7 days.” Same temperature?

Reply: Sorry about the confusion. “5 ” has been revised into “25 ”. 

“One tube of nematodes was used to extract the RNA and synthesize that RNA into cDNA directly without treatment (no treated group).” Day 0?

Reply: Sorry about the confusion. Actually, day 0 group was not used in this section. This is an incorrect description. We have removed this sentence. 

2.7 Mixed-stages Bx: which ones? How do you calculate survival rate?

Reply: Thanks. We have added more details about Mixed-stages Bx and survival rate calculation. 

Results:

3.1 Too succinct: “we performed RNA sequencing” of what? Please, describe the samples, the mapping step, include the percentage of mapping in Table 1 and the number of transcripts detected for each sample (indicate somewhere the total number of transcript for the Bx reference)

Reply: Thanks. The reference genome has 17,704 transcripts in total. This number together with other mapping details has been added in the material and methods section. We also added the percentage of mapping for each sample in revised Table 1. 

3.2 GO enrichment is not very informative (too general), could you describe some DEG (just to highlight 2 or 3 categories)

Reply: Thanks. We added some descriptions about 3 categories of DEGs including cytochrome P450s, HSPs and GPCRs. We also added more details to illustrate the reason why we chose these 6 BxGPCRs for further study in this section and 3.4. 

3.3 Figure 3: please, indicate in the legend the (putative) function of each gene

Reply: Thanks. The putative function of each gene has been added in the legend.

3.4 Add a figure showing alignment of the deduced amino acid sequences from the 6 predicted BxGPCR, and a known GPCR (from C. elegans), indicate the 7 transmembrane domains.

What are the percentage of identity between the 6 BxGPCR and with other GPCRs?

For QPCR analyses, indicate that the gene expression culminates at 3 days. 

Reply: Thanks. We have added the alignment of 6 predicted BxGPCR along with other known GPCR sequences we abstained from NCBI (See Figure S2). We also added transmembrane domains prediction in Figure S1. The gene expression culminates at 3 days has been indicated in QPCR section.

3.5 What are the homologies between the 6 BxGPCR nucleic sequences? I wonder if a dsRNA of 350 b is specific or can target another GPCR? Did you test the expression level of other GPCRs?

“The BxGPCR17454 RNAi was potent and specific for B. xylophilus.” What does it mean?

 Reply: Thanks. We have added the alignment of 6 BxGPCRs nucleic sequences in the revised manuscript (Figure S3). As illustrated in the new figure, the RNAi primer sequence is unique for BxGPCR17454. The QPCR results indicate that the primer can sufficiently dececase the transcript level of BxGPCR17454. So, these results can prove that this 350bp is unique and effective for the RNAi of BxGPCR17454. Sorry about the confusion about “The BxGPCR17454 RNAi was potent and specific for B. xylophilus.” , we have revise this into “The BxGPCR17454 RNAi was potent and specific in this study.” .

Discussion

-“We proved this theory in B. xylophilus through transcriptome analysis” too affirmative!

-Please include a brief description of GPCR: structure, how many genes have been described in C. elegans, …

-discussion on other DEG? HSP or other genes known to be related to thermal stress?

Reply: Thank you so much for these advises. We have carefully revised this part to describe our results in a less affirmative and not over estimated way. We revised the previous manuscript as following that our results can only prove an important role of BxGPCR17454not all the GPCRS in B. xylophilus. We also added a brief description of known GPCR in C. elegansin the introduction section. Other DEGs likewise HSP and P450 were also discussed in the revised manuscript. 

Reviewer 2 Report

I am of the opinion that while the Transcriptome-based analysis does reveal the regulation of BxGPCRs in Low Temperature Response, I believe it would be a bit of an overstatement to title the paper - "Crucial Role of BxGPCRs" especially because the confirmatory RNAi experiment was done on 1 BxGPCRs gene only.

As for the overall manuscript, I find it lacking a lot of details and explanations that typically accompany each experiment. Here are some cases in the Materials and Methods Section -

2.2

69 Product number of Oligo(dT) beads, how was the process carried out?

69 Fragmentation buffer : Which buffer was used? Was it made in the lab or commercial?

70-71 Reverse transcript into cDNA, DNA polymerase : Which enzyme, which products were used, how was it done?

72 : Ligated to Illumina sequencing adapters - how?

73 : PCR amplified : How many cycles?

2.3

76 What tool was used to remove adapters and reads with unknown nucleotides?

78 Were default parameters used for Bowtie2?

84 Were the RSEM results fed into edgeR as is, or after normalization into FKPM? This is very important and should be mentioned. The flow of data in the analysis is not very clear

86-87 How was the GO analysis carried out?

Similar details need to be given for ALL sections in Materials and Methods. It helps the reader understand clearly how each step was undertaken and leaves no scope for guesswork.

3Results

3.1 It would help to describe the clustering method and tool used. Likewise for correlation

3.2 In this section, the authors could introduce GPCRs. Why were they chosen? Where they show up in the GO analysis. Maybe a separate description or plot. 

Considering the entire paper is based on the 'crucial role of GPCRs', there is no introduction of what they are, and how the authors came to choose them for further analysis, over any other set of genes or pathways. This is highly unsettling to a reader, where they do not realize where the main target being studied came from, if there is no explanation or introduction.

Figure4A : Not clear on the relevance of this analysis and tree to the main story. As a general rule of thumb, the authors should talk about the reason behind doing each experiment/analysis and follow it up with a) What they observed, and b) How it contributed to the main story of the paper

Legend: the word 'label' is spelt two different ways in the description

3.5

The authors could explain the meaning of 'CK' a bit clearly. Earlier in the manuscript, it is mentioned that CK group refers to the 25C temperature samples. In this section however, the CK group seems like the group that was dsRNA-free. This is confusing, since it was not articulated clearly what CK means, and refers to.

208 : The results do indicate that the RNAi was potent and worked for BxGPCR17454. How does it mean that it was specific for B. xylophilus?

Figure 5C :The authors could elucidate a bit more on the results observed in the first 15 days of the Survival rate experiment. Would you expect both treated and untreated nematodes to have a similar survival rate for the first 15 days? Why?

Discussion

229 : It is overstating to write that the authors proved that the low temperature response process is regulated by many genes. The proof was actually done on 1 gene and the rest of the manuscript shows no other relation or pathway description or even an abundance heat map for just the GPCRs, that could hint towards the role of multiple genes in this system. While this is a step in the right direction, I feel it is hard to say at this point, with the current data that all the GPCRs play a crucial role in low temperature response process. It is possible that the authors do have enough data and evidence to back it up, but it does not show in the manuscript, due to the lack of plots and text to show connection between the GPCRs genes and their corresponding pathways in this system. 

Hence, the lines 255-258 are not entirely accurate in mentioning that functional validation was done on BxGPCRs gene's'. Moreover, no mention of the importance and role of G protein signaling has been described in the introduction, to direct the readers towards this pathway as opposed to any other.

Other edits :

13-14 : Grammatical restructuring needed. Possibly - "The causal agent of pine wilt disease is the pine wood nematode (PWN) (Bursaphelenchus xylophilus), whose ability to adapt different ecological niches is a crucial determinant of their invasion to colder regions.

17 : Spelling of "Sequencing"

31 : As one of the most dangerous plant "pests" in the world

39 : with the adults

40 : "fat accumulated fat content" ?

45 : Remove "etc" or replace with "among others"

46 : Some efforts have been made in the "study" of ?

50 : Could explain mix-staged (optional)

50 : performed RNA sequencing "on" mixed-stage PWN

52 : Novel GPCRs - Unclear how these are novel, not mentioned in the manuscript

53 : response pattern to low temperature

99 : incomplete conserve domain "were" removed.

101 : "Conserved" domains were analyzed

172 : 12 DEGs were selected randomly for qRT-PCR validation.

205 : "significantly" silenced

233 : is essential to

235 : reflect the impact of low temperature

264 Table S3 : The title says S4 in the actual spreadsheet

Author Response

Dear Reviewer,

Thank you so much for the reviewers’ comments concerning our manuscript. Those comments are all valuable and very helpful for revising and improving our paper, as well as the important guiding significance to our research. We have studied comments carefully and have made correction. We tried our best to improve the manuscript and made some changes in the manuscript. Revised portion are marked with "Track Changes" function in Microsoft Word in the paper.

With the help of your kind comments, we now understand that only one gene is not enough to prove “the crucial role of BxGPCRs”. So, we revised the title and paper topic have been revised into “Transcriptome-based Analysis Reveals a Crucial Role of BxGPCR17454in Low Temperature Response of Pine Wood Nematode (Bursaphelenchus xylophilus)”. 

We appreciate for reviewers’ warm work earnestly, and hope that the correction will meet with approval. Once again, thank you very much for your comments and suggestions.

Best regards,

Bowen Wang

Author's Reply to the Review Report (Reviewer 2)

2.2

69 Product number of Oligo(dT) beads, how was the process carried out?

Reply:Thanks. We have added more details about the process in the revised manuscript. 

69 Fragmentation buffer: Which buffer was used? Was it made in the lab or commercial?

Reply: Thanks. Full name of fragmentation buffer has been added in the manuscript. It is a commercial Kit by New England Biolabs. We have also added this in the manuscript.

70-71 Reverse transcript into cDNA, DNA polymerase : Which enzyme, which products were used, how was it done?

Reply: Thanks. Full name of DNA polymerase and other details about reverse transcription procedure have been added in the revised manuscript.

72 : Ligated to Illumina sequencing adapters - how?

Reply: Thanks. We have added details about ligation steps in the revised manuscript. 

73 : PCR amplified : How many cycles?

Reply: Thanks. PCR procedure was performed with 12-15 cycles. We have added this in the revised manuscript. 

2.3

76 What tool was used to remove adapters and reads with unknown nucleotides?

Reply: Thanks. fastq(version 0.18.0) was used here to remove adapters and reads with unknown nucleotides. We have added this in the revised manuscript. 

78 Were default parameters used for Bowtie2?

Reply: Thanks. Yes, default parameters were used here. We have added this in the revised manuscript. 

84 Were the RSEM results fed into edgeR as is, or after normalization into FKPM? This is very important and should be mentioned. The flow of data in the analysis is not very clear.

Reply: Sorry about the confusion. We use RSEM software to calculate the gene expression level which was normalized by using FPKM method. The edgeR package was then used to identify differentially expressed genes with FPKM across two groups. We have revised this in the manuscript.

86-87 How was the GO analysis carried out?

Similar details need to be given for ALL sections in Materials and Methods. It helps the reader understand clearly how each step was undertaken and leaves no scope for guesswork.

Reply: Sorry for the lack of details. All DEGs were mapped to GO terms in the Gene Ontology database (http://www.geneontology.org/), gene numbers were calculated for every term, significantly enriched GO terms in DEGs comparing to the genome background were defined by hypergeometric test. We have also added the method of Clustering, Pearson correlation, volcano plot, heatmap and GO categories analysis.

3Results

3.1 It would help to describe the clustering method and tool used. Likewise for correlation

Reply:Thanks. We have added the method of Clustering and Pearson correlation analysis in the Materials and Methods section.

3.2 In this section, the authors could introduce GPCRs. Why were they chosen? Where they show up in the GO analysis. Maybe a separate description or plot. 

Considering the entire paper is based on the 'crucial role of GPCRs', there is no introduction of what they are, and how the authors came to choose them for further analysis, over any other set of genes or pathways. This is highly unsettling to a reader, where they do not realize where the main target being studied came from, if there is no explanation or introduction.

Reply: Thanks. We added some descriptions about 3 categories of DEGs including cytochrome P450s, HSPs and GPCRs. We also added some descriptions, supplementary figure and supplementary table to illustrate the reason why we chose BxGPCRs for further study in the beginning of 3.4 section.

Figure4A : Not clear on the relevance of this analysis and tree to the main story. As a general rule of thumb, the authors should talk about the reason behind doing each experiment/analysis and follow it up with a) What they observed, and b) How it contributed to the main story of the paper

Legend: the word 'label' is spelt two different ways in the description

Reply: Sorry about the confusion. Details have been added in this section. We also added one supplementary table and two supplementary figures to illustrate the details about the reason why we chose these 6 BxGPCRs for further study. The legend has been revised. 

3.5

The authors could explain the meaning of 'CK' a bit clearly. Earlier in the manuscript, it is mentioned that CK group refers to the 25C temperature samples. In this section however, the CK group seems like the group that was dsRNA-free. This is confusing, since it was not articulated clearly what CK means, and refers to.

Reply: Sorry about the confusion. We have re-defined “CK” throughout the manuscript. CK group now is only refers to the 25C temperature samples. “CK” in Section 3.5 along with Figure 5 have been revised as “dsRNA-free group”.

208 : The results do indicate that the RNAi was potent and worked for BxGPCR17454. How does it mean that it was specific for B. xylophilus?

Reply:Sorry about the confusion about “The BxGPCR17454 RNAi was potent and specific for B. xylophilus.” , we have revise this into “The BxGPCR17454 RNAi was potent and specific in this study.” .

Figure 5C :The authors could elucidate a bit more on the results observed in the first 15 days of the Survival rate experiment. Would you expect both treated and untreated nematodes to have a similar survival rate for the first 15 days? Why?

Reply: Thanks.We did expect that treated and untreated groups have similar survival rate in the first 15 days. We have added following discussions in this section: Although the transcript level of BxGPCR17454 was evaluated in the first 7 days according to the qRT-PCR results, there are no differences between the survival rate of dsRNA-free group and dsRNA-treated groups in first 15 days. This may because that there are other downstream genes are influenced by the RNAi of BxGPCR17454. All these genes together with BxGPCR17454 influenced the survival rate of B. xylophilus under low temperature. Further studies need to be carried out to detect the downstream genes of BxGPCR17454 in the future.

Discussion

229 : It is overstating to write that the authors proved that the low temperature response process is regulated by many genes. The proof was actually done on 1 gene and the rest of the manuscript shows no other relation or pathway description or even an abundance heat map for just the GPCRs, that could hint towards the role of multiple genes in this system. While this is a step in the right direction, I feel it is hard to say at this point, with the current data that all the GPCRs play a crucial role in low temperature response process. It is possible that the authors do have enough data and evidence to back it up, but it does not show in the manuscript, due to the lack of plots and text to show connection between the GPCRs genes and their corresponding pathways in this system. 

Hence, the lines 255-258 are not entirely accurate in mentioning that functional validation was done on BxGPCRs gene's'. Moreover, no mention of the importance and role of G protein signaling has been described in the introduction, to direct the readers towards this pathway as opposed to any other.

Reply: Thank you so much for these advises. We have carefully revised this part to describe our results in a less affirmative and not over estimated way. We revised the previous manuscript as following that our results can only prove an important role of BxGPCR17454not all the GPCRS in B. xylophilus. We also added a brief description of known GPCR in C. elegansin the introduction section.

Other edits :

13-14 : Grammatical restructuring needed. Possibly - "The causal agent of pine wilt disease is the pine wood nematode (PWN) (Bursaphelenchus xylophilus), whose ability to adapt different ecological niches is a crucial determinant of their invasion to colder regions.

Reply: Thank you so much for this. The sentence has been revised as your suggestion. 

17 : Spelling of "Sequencing"

Reply: Sorry. The spelling has been revised to “Sequencing”

31 : As one of the most dangerous plant "pests" in the world

Reply: Sorry. “pest” has been revised to “pests

39 : with the adults

Reply: Thanks. “the” has been added 

40 : "fat accumulated fat content" ?

Reply: Sorry. This has been revised to “accumulated fat content”

45 : Remove "etc" or replace with "among others"

Reply: Thanks. We have replaced “etc” with “among others”

46 : Some efforts have been made in the "study" of ?

Reply: Thanks. This sentence has been revised to “Some efforts have been made in the study of…”

50 : Could explain mix-staged (optional)

Reply: Thanks. “mix-staged” has been revised to “mixed-stage” and further illustrated in detail in Material and Methods2.7.

50 : performed RNA sequencing "on" mixed-stage PWN

Reply: Thanks. “on” has been added in the sentence. 

52 : Novel GPCRs - Unclear how these are novel, not mentioned in the manuscript

Reply: Sorry. “Novel” has been removed from the sentence

53 : response pattern to low temperature

Reply: Thanks. “the” has been removed.

99 : incomplete conserve domain "were" removed.

Reply: Thanks. “was” has been revised to “were”

101 : "Conserved" domains were analysed

Reply: Sorry. “Conserves” has been revised to “Conserved”

172 : 12 DEGs were selected randomly for qRT-PCR validation.

Reply: Thanks. The sentence has been revised to “12 DEGs were selected randomly for qRT-PCR validation.”

205 : "significantly" silenced

Reply: Sorry. “significant” has been revised to “significantly”

233 : is essential to

Reply: Sorry. “are” has been revised to “is”.

235 : reflect the impact of low temperature

Reply: Thanks. Second “the” has been removed

264 Table S3 : The title says S4 in the actual spreadsheet

Reply: Sorry. The title of the actual spreadsheet has been revised to “S3”

Round 2

Reviewer 1 Report

My comments have been addressed appropriately. Just two remarks:

-In the M&M section, l 96, remove “respectively”

-In the results section “The proportion of total reads that mapped to the reference genome ranged from 69.99% 167 to 75.15%”. The values indicated in Table 1 (Mapping Rate) are different: why?

Author Response

Dear Reviewer,

Thank you so much for your kind approval of our Round 1 revision. We also thank you so much for the new comments concerning our manuscript. We have studied comments carefully and have made minor revisions. Revised portion are marked with "Track Changes" function in Microsoft Word in the paper.

We appreciate for reviewers’ warm work earnestly, and hope that the new correction will meet with approval. Once again, thank you very much for your comments and suggestions.

Best regards,

Bowen Wang

Author's Reply to the Review Report (Reviewer 1)

-In the M&M section, l 96, remove “respectively”

Reply:Sorry about the confusion. We have removed this “respectively” in M&M section. 

-In the results section “The proportion of total reads that mapped to the reference genome ranged from 69.99% 167 to 75.15%”. The values indicated in Table 1 (Mapping Rate) are different: why?

Reply: Sorry about the confusion. We are so carelessly at the very first that we confused the mapping rate with other parameters. We have now revised “from 69.99% to 75.15%” into “from 70.47% to 75.61%”. Thank you so much.

Reviewer 2 Report

The authors mention using FPKM counts for Differential Expression analysis using edgeR.

According to the statistical principles that tools like edgeR and DESeq are based on, they require raw counts as input. Feeding FPKM or other normalized values would be incorrect to the model that they operate on. I encourage the authors to read the edgeR manual or other articles online discussing the topic. Here are two examples -
https://www.reddit.com/r/bioinformatics/comments/3bx3em/fpkm_vs_raw_read_count_for_differential/

http://genomicsclass.github.io/book/pages/rnaseq_gene_level.html

As the authors have other orthogonal analyses to validate their results, I doubt that there will be major changes to the results. However, using normalized reads as input to edgeR is fundamentally incorrect, and should be fixed by using raw counts from RSEM as input.

FPKM and other normalized counts are used while making heat maps and clustering, but not for differential expression.

Other than that, the revised manuscript looks good!

Author Response

Dear Reviewer,

Thank you so much for your kind approval of our Round 1 revision. We also thank you so much for the new comments concerning our manuscript. We have studied comments carefully and have made minor revisions. Revised portion are marked with "Track Changes" function in Microsoft Word in the paper.

We appreciate for reviewers’ warm work earnestly, and hope that the new correction will meet with approval. Once again, thank you very much for your comments and suggestions.

Best regards,

Bowen Wang

Author's Reply to the Review Report (Reviewer 2)

The authors mention using FPKM counts for Differential Expression analysis using edgeR.

According to the statistical principles that tools like edgeR and DESeq are based on, they require raw counts as input. Feeding FPKM or other normalized values would be incorrect to the model that they operate on. I encourage the authors to read the edgeR manual or other articles online discussing the topic. Here are two examples –

https://www.reddit.com/r/bioinformatics/comments/3bx3em/fpkm_vs_raw_read_count_for_differential/

http://genomicsclass.github.io/book/pages/rnaseq_gene_level.html

As the authors have other orthogonal analyses to validate their results, I doubt that there will be major changes to the results. However, using normalized reads as input to edgeR is fundamentally incorrect, and should be fixed by using raw counts from RSEM as input.

FPKM and other normalized counts are used while making heat maps and clustering, but not for differential expression.

Reply: Sorry about the confusion. We have checked the details with the technician who did this part of work. We are so sorry for our misunderstanding about the details at the very first. Our study was carried out as you said in your kind comments. The “raw counts from RSEM”, not “FPKM”, were used as input in this part. We have now revised this in the revised manuscript. Thank you very much! 

Round 3

Reviewer 2 Report

Great!

Thank you for patiently addressing all the concerns and corrections